# Effects of Additives Containing Cyanopyridine on Electrodeposition of Bright Al Coatings from AlCl₃-EMIC Ionic Liquids

**Min Zhang [1], Dong Peng [1], Feifei Peng [1], Anwei Huang [1], Kaiqiang Song [1], Qingbing He [1], Changqing Yin [2], Jinsong Rao [2], Yuxin Zhang [2], Haitao Chen [1], Dalong Cong [1,*] and Zhongsheng Li [1,*]**

[1] Southwest Institute of Technology and Engineering, Chongqing 401329, China; zhmcq59@163.com (M.Z.); pd2019@126.com (D.P.); sunshare100@tom.com (F.P.); huanganwei12@163.com (A.H.); scut_song@163.com (K.S.); heqingbing@sina.com (Q.H.); chenhaitao5196@163.com (H.C.)

[2] College of Material Science and Engineering, Chongqing University, Chongqing 400044, China; CorsacY@163.com (C.Y.); rjs@cqu.edu.cn (J.R.); zhangyuxin@cqu.edu.cn (Y.Z.)

* Correspondence: congdl09@163.com (D.C.); zhongshli@163.com (Z.L.); Tel.: +86-23-68792314 (D.C.)

**Abstract:** The Al coatings were electrodeposited on the Cu substrate from AlCl₃-EMIC ionic liquid (AlCl₃:EMIC = 2:1 molar ratio) containing three cyanopyridine additives with different positions of the substituent group on the pyridine ring, which were 2-cyanopyridine, 3-cyanopyridine, and 4-cyanopyridine. The effects of cyanopyridine additives on the deposition potential, morphology, brightness, and corrosion properties of Al coatings were investigated. It was considered that the deposition potential of Al shifted to more negative overpotentials, the quality of Al coatings was promoted, and the corrosion property was improved by the cyanopyridine additives to a varying degree. Especially in the presence of 4-cyanopyridine, the flattest mirror bright Al coating was obtained, which had the smallest homogeneous nanocrystal grain size and strongest Al (200) crystallographic orientation. The average roughness Ra value was as low as 31 nm compared to that in the absence of cyanopyridine additives, which was 417 nm. Furthermore, the corrosion current density of the bright Al coating was three orders of magnitude lower than the rough Al coating, which resulted from the dense nanocrystal structure.

**Keywords:** bright Al; electrodeposition; ionic liquids; cyanopyridine; additives

## 1. Introduction

Ionic liquids are a new kind of green solvent that allow the electrodeposition of Al and many other water-sensitive metals because of their outstanding properties, such as wide electrochemical potential windows, nonvolatility, and high thermal and electrochemical stability [1–4]. Especially, electrodeposited Al and Al alloy coatings from ionic liquids have received considerable attention because of their excellent physicochemical properties in the past decades, which not only had low density and good strength, but also possessed outstanding corrosion resistance [5–8]. Compared to various other available coating techniques, electrodeposition is such a widely applied, low-cost, and highly developed technology that it permits the coating of complex geometries homogeneously, offers high-purity deposits (≥99.99% Al), and allows for the deposition of thick layers at room temperature [9–12].

As we all know, the microstructure of the metal coatings has a remarkable impact on its properties, such as brightness and corrosion resistance. Numerous studies have been published to modulate the microstructure of the metal coatings to improve their performance by adding appropriate alloying elements [13–15], changing the temperature, and optimizing the current density [16–18]. Although Al coatings can be electroplated from the neat ionic liquids, the surface morphology of the electrodeposit is usually rough and

gray. The use of appropriate additives into the electrolyte is one of the key technologies to regulate the microstructure and enhance the quality of the Al electrodeposit [19,20].

In the presence of brightener, a mirror bright aluminum coating can be directly obtained, which not only increases the surface quality but also enhances the corrosion resistance and other performances [2,8,21]. Brightened agents can be divided into two sorts according to the different chemical components, inorganic chloride and organic molecules. For the inorganic chloride additives, researchers have mainly focused on the study of alkali metal chloride (such as LiCl and NaCl) [22,23] and rare earth chloride (such as $LaCl_3$ and $CeCl_3$) [24] in the ionic liquid electrodeposition of Al coatings. Liu et al. [22] added LiCl and NaCl to $2AlCl_3$-BmimCl to obtain nano-aluminum grains. Abbott et al. [23] found that the additive LiCl in ionic liquid was conducive to the formation of a dark gray Al layer with large particle size, and it was difficult to obtain a leveling effect. Li et al. [24] found that the addition of $LaCl_3$ was helpful to obtain a bright and flat surface of the Al coating in the $AlCl_3$-EMIC system. For the organic molecule additives, researchers mainly paid attention to the organic small molecules [25], such as benzene [4,26], nicotinamide [27], nicotinic acid [20], methyl nicotinate, light aryl naphtha [28], ethylene glycol [29], 2-chloronicotinyl chloride [30], and phenanthrene derivatives [21,31]. Endres et al. [20] found that uniform and nanocrystalline Al coatings can be electrodeposited from the ionic liquid [Emim]Cl/$AlCl_3$ with the addition of nicotinic acid. Caporali et al. [21] found that a bright Al coating could be obtained from the ionic liquid [Bmim]Cl/$AlCl_3$ with 1, 10-phenanthroline as a brightener. At the same time, Zhang et al. [27] investigated the Al deposition in BMIC/$AlCl_3$ with niacinamide as an additive, and they obtained highly uniform and smooth Al deposits with an average crystalline size of 14 nm.

Completely, a proper additive has a significant effect on the electrodeposition of Al from ionic liquid, and it is very important to know how to choose effective additives and how different additives affect the quality of the aluminum deposition layer obtained from ionic liquids. It is worth noticing that pyridine derivatives have excellent effects as additives in ionic liquid baths due to their stronger polarization ability [4,23,26], and on this basis, people have investigated the influence of different structures and positions of the functional groups on the morphology and properties of Al coatings electrodeposited from ionic liquid. Leng et al. [32] studied the effect of organic molecular additives containing hydroxyl functional groups on the Al deposition layer, and they found that the addition of methyl carbamate significantly improves the brightness and compactness of the Al layer due to the adsorption on the cathode surface. Wang et al. [33] demonstrated that nicotinic acid and methyl nicotinate, which had electron-withdrawing substituent groups on the pyridine ring, can be absorbed on the electrode surface more easily than the 3-methyl pyridine, which has an electron-donating substituent group, and thus served as very effective brighteners producing highly uniform and smooth Al coatings from 1-butyl-3-ethylimidazolium-$AlCl_3$ baths. Yamazaki et al. [34] demonstrated that the addition of 4-pyridinecarboxylic acid hydrazide to an EMIC/$AlCl_3$/toluene mixture improves the brightness of the deposited Al film, and the structural isomers with the same electron-absorbing groups brought about different surface brightnesses.

According to the recent research, pyridine derivatives have excellent effects as additives in $AlCl_3$-EMIC ionic liquid baths due to their stronger polarization ability, and on this basis, cyanopyridine is expected to become an excellent additive improving the quality of Al coatings because of its pyridine structure and strong electron-withdrawing substituents. Previously, we had discussed the effect of the different concentrations of 4-cyanopyridine additives on the quality of the Al-Mn coatings electrodeposited in EMIC-$AlCl_3$-$MnCl_2$ baths [35]. Compared with the other various additives, the brightness of Al coatings was obviously improved with a small concentration (6 mM) of 4-cyanopyridine. Nevertheless, the relation between the position of cyanogen substituent groups and surface brightness of Al coatings has not been illustrated yet. In this study, we selected three cyanopyridine additives, which were 2-cyanopyridine, 3-cyanopyridine, and 4-cyanopyridine, and inves-

tigated their effect on the brightness and properties of Al deposits from AlCl$_3$-EMIC ionic liquid (AlCl$_3$:EMIC = 2:1 molar ratio).

## 2. Experimental

### 2.1. Materials

An industrial pure Cu sheet was cut into 40 mm × 10 mm × 1 mm-sized disks for the substrate. High-purity Al plate and Al wire (99.999%) were used for the anode electrode and for refining ionic liquids. The chemical materials involved in this experiment were anhydrous AlCl$_3$ (Alfa Aesar, Tianjin, China, AR), 2-cyanopyridine (Aladdin, Shanghai, China, AR), 3-cyanopyridine (Aladdin, Shanghai, China, AR), and 4-cyanopyridine (Aladdin, Shanghai, China, AR). The molecular structure of the three cyanopyridine additives is shown in Figure 1. The 1-ethyl-3-methylim-idazoliumchloride (EMIC) was synthesized according to [20] experimentally.

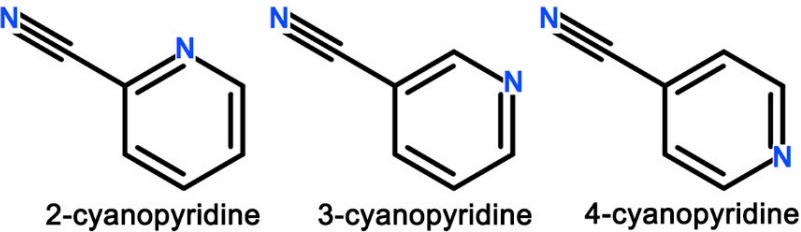

**Figure 1.** Molecular structure diagrams of the three cyanopyridine additives.

### 2.2. Preparation of Ionic Liquid Electrolyte

In a glove box (SG1200/1000TS, Vigor Co., Chengdu, China) with low oxygen and water content (<1 ppm) protected by argon gas, AlCl$_3$ was slowly added to the EMIC (AlCl$_3$:EMIC = 2:1 molar ratio), mixed well, and then poured into a jar with high-purity Al wires for purification at room temperature. The three additives (2-cyanopyridine, 3-cyanopyridine, and 4-cyanopyridine) were measured at the concentration of 6 mM and then added into the refined AlCl$_3$-EMIC ionic liquid. Most organic additives are toxic, compared to the various other additives; the brightness of the Al coating obviously improved with a small concentration (6 mM) of 4-cyanopyridine. Although the cyanopyridine additives are harmful when inhaled and contact the skin, the glove box and protective clothing avoid the contact during the test.

### 2.3. Pretreatment

The Cu substrate was polished with 4000# sandpaper successively for removing the surface oxide layer, and then ultrasonically cleaned for 10 min in deionized water and anhydrous ethanol. The apparent area for electrodeposition was 10 mm × 10 mm, and the other portion was pasted and covered with insulated tape. The high-purity Al plates were ultrasonically cleaned by NaOH, H$_3$PO$_4$, deionized water, and anhydrous ethanol for 15 min, and then dried by air. The cleaned Cu and Al plates were placed in the glove box at once.

### 2.4. Electrodeposition of Al Coatings

Al coatings were electrodeposited from AlCl$_3$-EMIC ionic liquid with or without additives at a current density of 10 mA/cm$^2$ for 1 min, 10 min, and 30 min in a glove box protected by argon gas. After the electrodeposition, the sample was removed from the glove box and the residual ionic liquid was cleaned on the surface, immediately.

### 2.5. Characterization

The brightness of the coating was characterized by the reflection ability of the coating on the surface of the linear pattern. The surface morphology and composition of the coating was observed by an emission scanning electron microscope (SEM, JSM-7800F, JEOL, Tokyo,

Japan) with energy-dispersive spectroscopy (EDS, INCA Energy 350 Oxford, Oxford, UK). The phase structure of the coating was analyzed by X-ray diffraction (XRD, Smartlab-9, RIGAKU Co., Tokyo, Japan). Surface roughness analysis was carried out using an atomic force microscope (AFM, Ntegtra Prima, NT-MDT, Moscow, Russia).

Electrochemical measurements were elucidated by an electrochemical workstation (Princeton Parstat 2273, ARAMTEK Co., PA, USA) with a three-electrode system. For the cyclic voltammetry (CV) test, the electrolyte was a 2:1 M ratio $AlCl_3$-EMIC ionic liquid containing none or various additives. The polished Cu plate was used for the working electrode, while the high-purity Al plate (99.999%) was regarded as the counter electrode, and the pure Al wire (99.999%) with a diameter of 2 mm placed in a separate fritted glass tube sealed by a porous ceramic mesh containing the 2:1 $AlCl_3$-EMIC ionic liquid was assembled as the reference electrode. The electrode potential was swept starting from the open-circuit potential (OCP) in the cathodic direction down to −0.8 V and then back to +1.2 V with a scan rate of 1 mV/s. The corrosion behaviors of the Cu substrate and Al coatings were evaluated by potentiodynamic polarization measurements in a 3.5 wt.% NaCl aqueous solution at a scan rate of 1 mV/s. A Pt sheet and a standard calomel electrode (SCE) were used as the counter electrode and the reference electrode, respectively.

## 3. Results and Discussion

### 3.1. Voltammogram Measurements

Figure 2 displays the voltammetry of the EMIC-$AlCl_3$ ionic liquid affected by the additives. It could be obviously seen that the additives had a symbolized impact on the voltametric responsive, suggesting the process of nucleation and growth affected by the additives [17,36]. With addition of 6 mM of 2-cyanopyridine, 3-cyanopyridine, and 4-cyanopyridine in the bath, the onset deposition potential of Al alloy underwent a negative migration, which was −70 mV, −90 mV, and −100 mV vs. Al, respectively. These implied that the deposition overpotential of Al increased and the electrochemical polarization improved. Corresponding to the promotion of Al nucleation on the cathode, the polarization ability was in the sequence of 4-cyanopyridine > 3-cyanopyridine > 2-cyanopyridine. At the same time, it was noted that the addition of various additives brought about the increase in the cathodic deposition peak current at position B (−300 to −480 mV), compared with neat ionic liquid. Moreover, the anodic sweep had two clear corresponding stripping peaks $A_1$ (−50 to 90 mV) and $B_1$ (270 to 380 mV) with each additive, and these had been regarded as Al grown in a two-step deposition process, nanocrystal and bulk material [23].

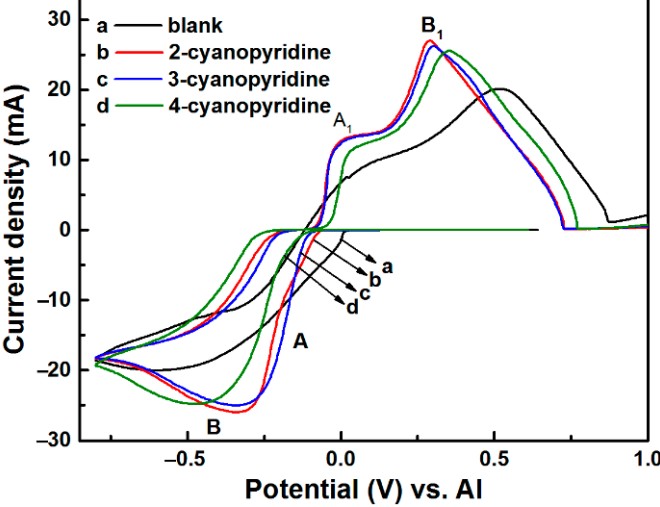

**Figure 2.** Cyclic voltammogram recorded on the Cu electrode in $AlCl_3$-EMIC ionic liquids containing (**a**) no and 6 mM of (**b**) 2-cyanopyridine, (**c**) 3-cyanopyridine, and (**d**) 4-cyanopyridine at 303 K.

### 3.2. Electrodeposition and Microstructures of Al Coatings

Figure 3 shows the macroscopic photos of AlCl$_3$-EMIC ionic liquids with and without various additives and deposited Al coatings from corresponding solutions. The appearance color for ionic liquids, the AlCl$_3$-EMIC ionic liquid without additive, was colorless and transparent (Figure 3a). After adding 2-cyanopyridine, the color of the ionic liquid became darker brown (Figure 3b), and after adding 3-cyanopyridine and 4-cyanopyridine, the appearance color changed slightly, which was light yellowish (Figure 3c,d). We speculated that the melting point of 2-cyanopyridine was 299–301 K, which was brown liquid at room temperature. When it was employed as an additive, it might change the property of the ionic liquid; however, this needs to be proved in a further study.

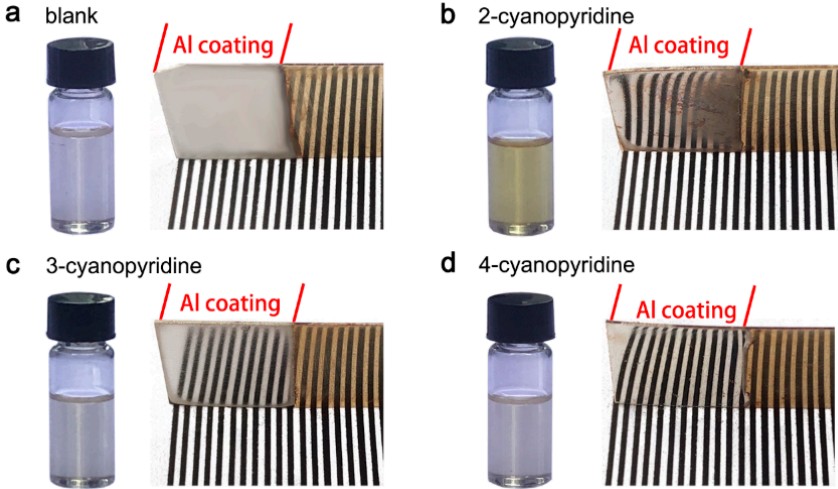

**Figure 3.** Appearance of AlCl$_3$-EMIC ionic liquids and photographs of Al coatings on the Cu substrate electrodeposited at 10 mA/cm$^2$ for 30 min from solutions containing (**a**) no and 6 mM of (**b**) 2-cyanopyridine, (**c**) 3-cyanopyridine, and (**d**) 4-cyanopyridine.

We took photos of samples on which the printed-out "linear patterns" were reflected. Then, the brightness of Al coatings deposited from AlCl$_3$-EMIC ionic liquids was compared. The sample (Figure 3a) obtained from pure AlCl$_3$-EMIC ionic liquids had a milky white and matte surface, which could not reflect the linear pattern in front of it. When 2-cyanopyridine was added, part of the coating (Figure 3b) reflected the linear pattern in front of the coating, but the coating edges appeared black. Furthermore, with the addition of 3-cyanopyridine or 4-cyanopyridine, the Al coatings (Figure 3c,d) gradually became brighter. Particularly, the sample was metallic-looking and bright with a mirror finish, which could reflect the linear pattern clearly in the presence of 4-cyanopyridine (Figure 3d). Above all, the cyanopyridine showed the effect of a brightener added into the AlCl$_3$-EMIC ionic liquids.

After comparing the differences in surface brightness, the microstructure of the coating was also systematically analyzed. As shown in Figure 4, the grain sizes of the samples were significantly smaller by adding the cyanopyridine additives than those that were prepared using neat electrolyte, and there were also significant differences in morphology. In the absence of additives, the aluminum particles were coarse and bulk where the heterogeneity grain size was 1.0–2.0 μm (Figure 4a), the thickness of the film was approximately 10.2 μm, and the edge of coating had obvious fluctuation (Figure 4b). After adding cyanopyridine additives, the SEM images of the surfaces showed very different morphologies to that without additives. For adding 2-cyanopyridine, the grain size decreased significantly from microscale to nanoscale, and parts of the nanoparticles agglomerated together to form a larger particle (Figure 4c). The film thickness was about 8.7 μm and the coating's edge became flatter (Figure 4d). For adding 3-cyanopyridine and 4-cyanopyridine, the independent nanoparticles were homogeneous with particle sizes of less than 100 nm (Figure 4e,g), and the film thicknesses decreased to 7.8 and 7.6 μm, respectively (Figure 4f,h).

Meanwhile, all the interfaces between the Al coatings and Cu substrates were seamless, indicating that the two were combined closely. When additives were included in the solution, the thickness of Al films obtained with the additives was a little lower than those without additives, which might result from the inhibiting growth of the cyanopyridine additives. Combined with Figures 2 and 3, it was evident that the higher the overpotential was, the smaller the grain size was, leading to a brighter surface.

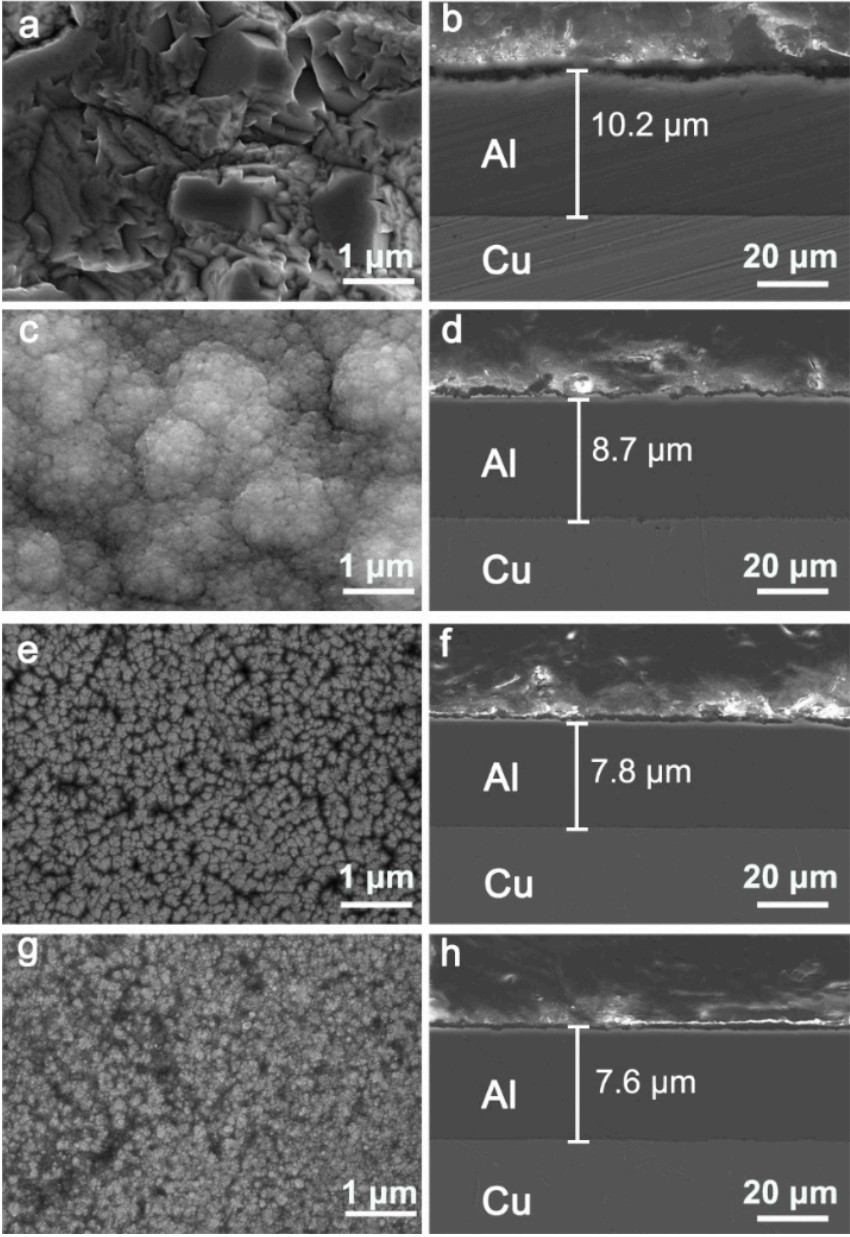

**Figure 4.** Surface and cross-sectional morphology of Al coatings on Cu substrate electrodeposition at 10 mA/cm$^2$ for 30 min from AlCl$_3$-EMIC ionic liquids containing (**a**,**b**) no and 6 mM of (**c**,**d**) 2-cyanopyridine, (**e**,**f**) 3-cyanopyridine, and (**g**,**h**) 4-cyanopyridine. (**a**,**c**,**e**,**g**) are the corresponding SEM surface images of coated samples at a 20,000× magnification, and (**b**,**d**,**f**,**h**) are the corresponding cross-sectional images of the coated samples at a 1000× magnification.

Besides, as shown in Figure 4b, a black area appeared on the Al coating in the presence of 2-cyanopyridine, and the corresponding SEM morphology is displayed in Figure 5. It was conspicuous that large and deep cracks formed and spread throughout the deposited Al coating. The EDS spectrum showed strong Al and O peaks without any other element.

The concentration of element O of point 2 (gray area) was much higher than that of point 1 (white area), which might be due to the cracks gathering more oxygen.

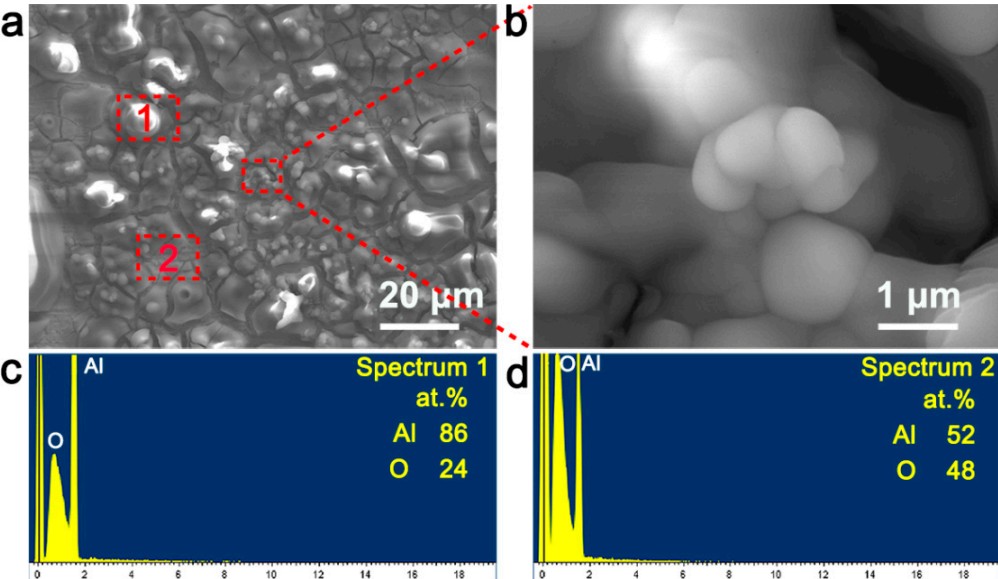

**Figure 5.** Surface morphology of black area in Figure 4b at magnitudes of (**a**) 1000× and (**b**) 20,000×, and (**c,d**) EDS spectrum of corresponding area of points 1 and 2.

In order to study the growing process of the Al coating with and without cyanopyridine additives, we electrodeposited Al coatings in 1 min and 10 min, respectively. As seen in Figure 6, when the electrodeposition time was 1 min, all the Al coatings displayed high brightness. Whereas the Al coating had a black area at the corner in the $AlCl_3$-EMIC ionic liquids with 2-cyanopyridine solvent, the other Al coatings were uniform. From the microstructure, the average grain size was 0.1–0.5 μm in the pure electrolyte and decreased to nanometer scale with additives of 2-cyanopyridine, 3-cyanopyridine, and 4-cyanopyridine. While the electrodeposition time extended to 10 min, the brightness of the Al coating (Figure 6b) obtained from pure $AlCl_3$-EMIC ionic liquids reduced a great deal and displayed a white coarse surface, which was due to the obvious growing grain size up to 0.5–1.5 μm. The brightness of the Al coating deposited from the $AlCl_3$-EMIC ionic liquids containing different cyanopyridine additives was still maintained. The whole edge of the Al coating (Figure 6d) became black, while the SEM photo showed cracks when 2-cyanopyridine was added to the $AlCl_3$-EMIC ionic liquid (Figure 6). When the electrodeposition time was further prolonged to 30 min, seen in Figure 4, there were large and small grains coexisting with the absence of additives, and the larger grain size of the Al coating gradually increased to 1.5–2 μm. With the presence of cyanopyridine additives, the grain size of Al deposits did not change significantly compared with the result of 10 min, and it was still nanoscale. In summary, the cyanopyridine additives showed a strong grain refinement effect on Al deposits from EMIC-$AlCl_3$ ionic liquid. In addition, a remarkable inhibition of the growth of deposited Al nuclei was obtained from ionic liquid containing 3-cyanopyridine and 4-cyanopyridine additives (Figure 4e,g and Figure 6f,h). Especially in the presence of 4-cyanopyridine, the Al coating had a better brightness with a smaller nanocrystal size.

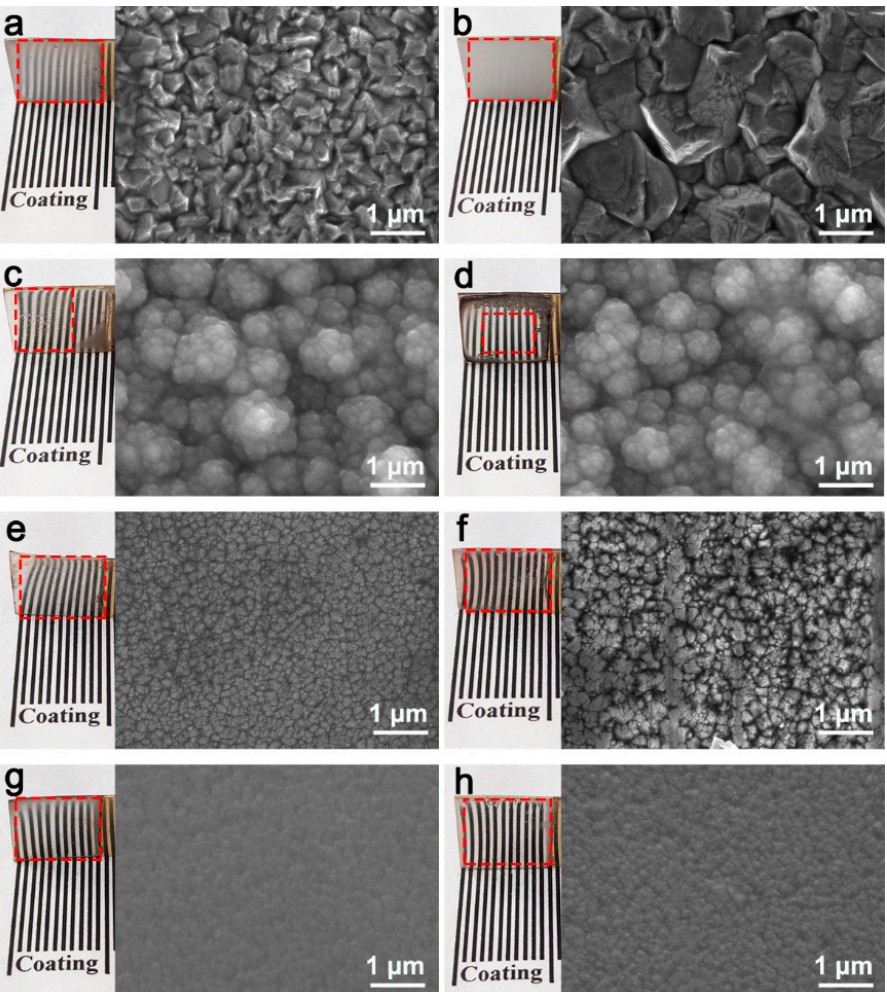

**Figure 6.** Optical images and SEM images of Al coatings obtained from AlCl$_3$-EMIC ionic liquids containing (**a,b**) no and 6 mM of (**c,d**) 2-cyanopyridine, (**e,f**) 3-cyanopyridine, and (**g,h**) 4-cyanopyridine, with a current density of 10 mA/cm$^2$ for (**a,c,e,g**) 1 min and (**b,d,f,h**) 10 min. The SEM images of the coated samples are at a 20,000× magnification.

Figure 7 illustrates the X-ray diffraction peaks of the Al coating and Cu substrate confirmed from XRD patterns. It can be seen that the four main (111), (200), (220), and (311) crystal structures dominated in the absence of additives, while the same (200) dominated with 6 mM of each additive, and the (111), (220), and (311) reflections were relatively weak or even invisible. The vanished (220) and (311) characteristic diffraction peaks of the coatings supported the idea that the additives are adsorbed at the electrode–solution interface, which changed the deposit morphology and nucleation mechanism. The texture coefficients ($TC_{(hkl)}$) of the (111), (200), (220), and (311) crystal planes were quantified by applying the following formula (Equation (1)) [37]:

$$TC_{(hkl)} = \frac{I_{hkl} / I_{r,hkl}}{1/n \sum I_{hkl} / I_{r,hkl}} \tag{1}$$

where $I_{hkl}$ is the peak intensity of the ($hkl$) crystal plane for the obtained Al coatings, $I_{r,hkl}$ is the peak intensity of the ($hkl$) crystal plane for the JCPDS card no. 04-0787, and $n$ is the total number of considered crystal planes. The results of the texture calculations for the various Al coatings obtained from the XRD patterns in Figure 7 are demonstrated in Table 1. The Al deposit obtained from the neat ionic liquid was preferentially textured, taking the orientation of the (220) crystal planes. The intensity of the (111) plane was basically equal to that of a randomly oriented sample, and the (220) and (311) planes

were relatively weak. Nevertheless, the deposits from the ionic liquids including various cyanopyridine additives had a strong preferred (200) reflection. A comparison of the (200) peaks of the various Al coatings revealed that the (200) peak of the Al-4-cyanopyridine coating was observably broadened, suggesting that the smaller grain sizes were obtained during the Al-4-cyanopyridine coating (Figure 7). The crystalline domain sizes of the Al deposits were calculated from the full-width at half-maximum of the (200) peak in the XRD profiles using the Sherrer equation. The average crystallite sizes of the deposits obtained with 3-cyanopyridine and 4-cyanopyridine additive were 32 and 24 nm, which were very consistent with the SEM micrographs in Figure 4. The XRD results demonstrated that cyanopyridine additives had an obvious effect on the preferred orientation of the Al (200) peaks and refining grains, and both of them led to the increasing brightness of the Al coatings.

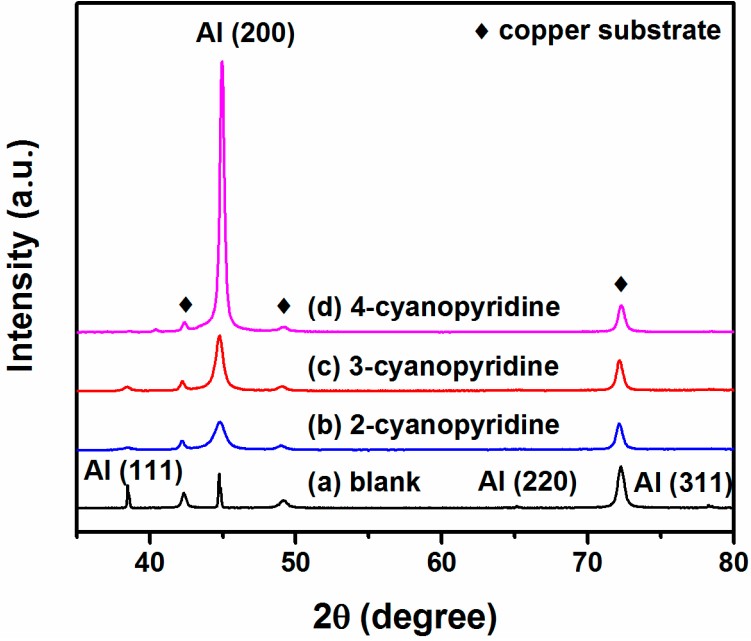

**Figure 7.** XRD patterns of Al coatings electrodeposited from AlCl$_3$-EMIC ionic liquids with (**a**) no and 6 mM of (**b**) 2-cyanopyridine, (**c**) 3-cyanopyridine, and (**d**) 4-cyanopyridine.

**Table 1.** Texture coefficients of the Al deposits obtained from the XRD patterns in Figure 7.

| TC (*hkl*) | TC (111) | TC (200) | TC (220) | TC (311) |
|---|---|---|---|---|
| Al-blank | 0.99 | 1.36 | 0.57 | 0.61 |
| Al-2-cyanopyridine | 0.48 | 2.38 | 0.0 | 0.0 |
| Al-3-cyanopyridine | 0.34 | 2.51 | 0.0 | 0.0 |
| Al-4-cyanopyridine | 0.0 | 2.84 | 0.0 | 0.0 |

Furthermore, AFM was adopted to test the roughness of Al coatings affected by the cyanopyridine additives. As shown in Figure 8, the crystal size of the Al surface was considerably smaller and more uniform with each additive than that from the neat electrolyte. The surface average roughness (Ra) was evaluated for the surface area of 100 μm × 100 μm. In the absence of additives, the average roughness Ra value was 417 nm, and the corresponding average values of Ra were 56, 35, and 31 nm for the samples containing different cyanopyridine amounts. The Ra results were due to the strong grain refinement and crystal plane preferential orientation effects of the cyanopyridine, confirmed in the SEM and XRD results.

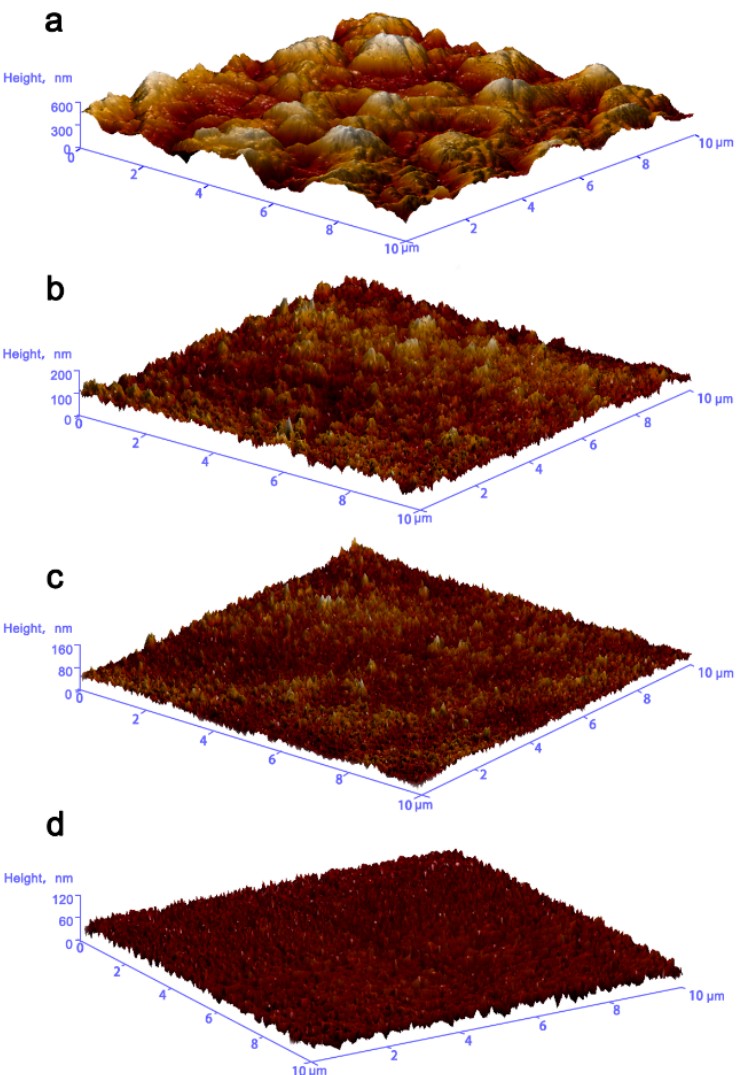

**Figure 8.** AFM images of Al coatings deposited at 10 mA/cm$^2$ for 30 min from AlCl$_3$-EMIC ionic liquids containing (**a**) no and 6 mM of (**b**) 2-cyanopyridine, (**c**) 3-cyanopyridine, and (**d**) 4-cyanopyridine.

In summary, the brightest Al coating was obtained from AlCl$_3$-EMIC ionic liquids containing 4-cyanopyridine, which acted as a brightener and leveling agent. This would result from the following reasons. First, the addition of 4-cyanopyridine made a significant overpotential required for Al electrodeposition, and the higher the overpotential was, the larger the cathode polarization and the greater the nucleation rate of Al were. The Al nuclei grew to become a dense and smooth coating more easily. Meanwhile, the activity of the crystal face was consistent and bright coatings could be obtained afterward. In this study, flat, shiny, and highly uniform Al layers with nanocrystals and a (200) preferential orientation were obtained with the 4-cyanopyridine additives. The light could be reflected as on a completely smooth surface, and the coating appeared bright then.

### 3.3. Corrosion Properties

The corrosion resistance of the Al-coatings was measured by potentiodynamic polarization curves in a 3.5 wt.% NaCl aqueous solution, shown in Figure 9, and Table 2 summarizes the Al samples' corrosion potential ($E_{corr}$) and corrosion current density ($I_{corr}$). The corrosion potential of the different Al alloy coating electrodes was −0.764 V, −0.703 V, −1.057 V, and −1.123 V. By comparing the polarization curve of the pure Al coating, the self-corrosion potential of the Al coating with 2-cyanopyridine moved slightly positive. At the same time, the self-corrosion potential of Al samples with 3-cyanopyridine and

4-cyanopyridine additive was much more negative to that of the pure Al coating. On the other hand, the pure Al coating had a large corrosion current density of $2.9 \times 10^{-5}$ A/cm$^2$. In the presence of cyanopyridine additives, the corrosion current density of Al deposited was much lower than that without additives. Particularly, when adding 3-cyanopyridine and 4-cyanopyridine, the corrosion current density characterized by much smaller values (respectively, $9.88 \times 10^{-8}$ and $6.34 \times 10^{-8}$ A/cm$^2$) reduced by three orders of magnitude than the corresponding values of Al-blank. This result was possibly due to the nanometer grain size and good combination with the Cu substrate (Figure 4). For the Al coating with 2-cyanopyridine, the cracks and nodular particles led to a higher corrosion current density ($1.02 \times 10^{-6}$ A/cm$^2$) than those with 3-cyanopyridine and 4-cyanopyridine. Even though the evaluation of the corrosion current could be conducted only qualitatively, these data in the paper showed that the bright Al coating, especially with 4-cyanopyridine, acted as an efficient anticorrosion barrier rather than the rough Al coating.

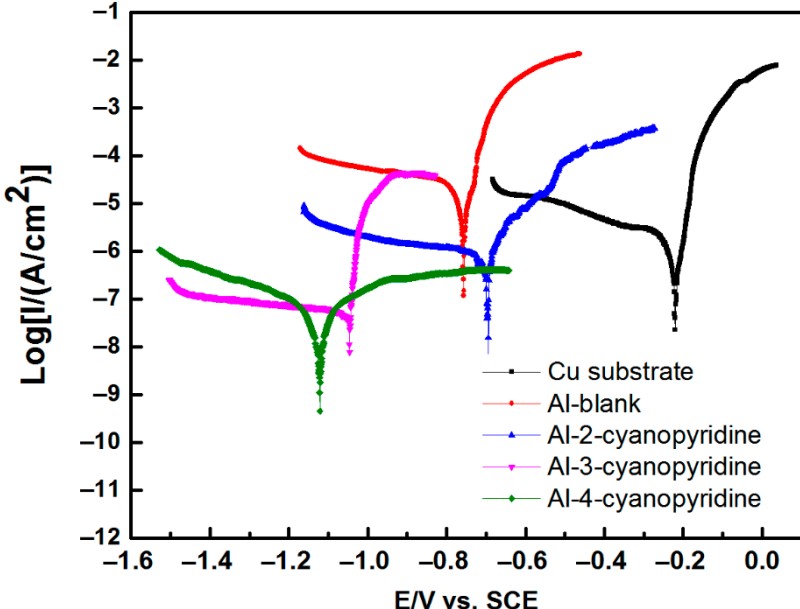

**Figure 9.** Polarization curves of different Al alloy coatings on Cu substrate in 3.5 wt.% NaCl solution at a scan rate of 1 mV/s.

**Table 2.** Corrosion parameters received from the polarization curves in Figure 9.

| Materials | $E_{corr}$ (V) | $I_{corr}$ (A/cm$^2$) |
|---|---|---|
| Al-blank | −0.764 | $2.9 \times 10^{-5}$ |
| Al-2-cyanopyridine | −0.703 | $1.02 \times 10^{-6}$ |
| Al-3-cyanopyridine | −1.057 | $9.88 \times 10^{-8}$ |
| Al-4-cyanopyridine | −1.123 | $6.34 \times 10^{-8}$ |

### 3.4. Discussion

From the above-mentioned results, when the three cyanopyridine additives were added in the AlCl$_3$-EMIC ionic liquids, the surface quality and corrosion resistance of the Al plating by electrodeposition were promoted. Among them, the promoted effect was best when adding 4-cyanopyridine, followed by 3-cyanopyridine and 2-cyanopyridine. As we know, the three additives were pyridine derivatives with the same type of cyanogen substituent groups, and the difference in position on the pyridine ring was comparable to the N atom (Figure 1). Cyanogen had a strong electron-absorbing group structure, which was beneficial to the adsorption of the additive on the cathode, and the ability was in the order of 4-cyanopyridine > 3-cyanopyridine > 2-cyanopyridine. The ability of adsorption led to the surface quality of Al, such as brightness, smoothness, density, and

grain size. From this study, the results of the experiment indirectly proved the difference in the adsorption capacity of different isomers of cyanopyridine and 4-cyanopyridine, which had an N atom in the aromatic ring and an acetyl hydrazine group located in the para position, producing the Al coating with a high reflectance among the isomers of cyanopyridine [34].

The phenomenon of grain refinement and directional growth of the Al coating caused by cyanopyridine additives would be due to the electrochemical polarization caused by adsorption. Figure 10 schematically expresses the growth mechanism for the Al coating deposited from neat AlCl$_3$-EMIC ionic liquid and the electrolyte with cyanopyridine. Due to the double layer structure of ionic liquids, especially the adsorption of particles in the tight layer, the electrodeposition process was significantly affected. Even a small amount of adsorption will largely affect the precipitation and location of metal on the cathode, and also affect the subsequent metal crystallization and compactness. Due to the greater electrochemical polarization caused by the adsorption of cyanopyridine, the nuclei that formed from the electrolyte with cyanopyridine were smaller and more than those in the blank electrolyte. Additionally, cyanopyridine additives adsorbed on the cathode can not only increase the nucleation rate but also adsorb on the original crystal plane, especially the growth point, and thus slow down the growth rate of the original crystal plane. These led to the preferentially oriented Al coatings with fine grains and a smooth surface. According to the current data, the cause of the edge effect of 2-cyanpyridine could not be fully revealed, and its mechanism and process must be further studied in future work.

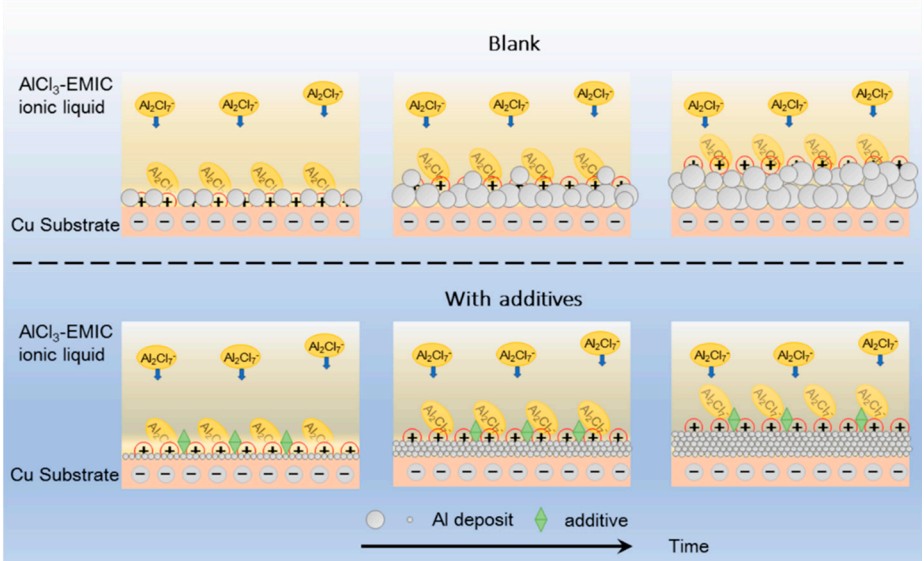

**Figure 10.** Schematic diagrams of the Al coating electrodeposition process from AlCl$_3$-EMIC ionic liquids containing no (blank) and additives.

## 4. Conclusions

The effects of cyanopyridine additives for the Al coating electrodeposited from AlCl$_3$-EMIC ionic liquid were investigated in this paper. The results showed that all the additives improved the quality of the Al coating, refined the grain size, and increased the corrosion resistance to varying degrees. With the addition of cyanopyridine additives, the reduction overpotential of Al was more negative than that of the neat AlCl$_3$-EMIC ionic liquid, and the effect on the deposition potential was in the order of 4-cyanopyridine > 3-cyanopyridine > 2-cyanopyridine. The 4-cyanopyridine acted as the most effective brightener and leveling agent, leading to the brightest mirror finish Al coating obtained. The grain size was as small as 24 nm and the surface roughness was as low as 31 nm with the addition of 4-cyanopyridine. In addition, the Al coating showed excellent corrosion properties, proved by the corrosion current density reducing by three

orders of magnitude. With the electrodeposition time extended from 1 min to 30 min, there was no obvious grain coarsening observed, demonstrating the effect of inhibiting grain growth by the cyanopyridine additives. The coatings were homogeneous and dense, composed of nanocrystalline particles, and had an obvious (200) preferential orientation.

**Author Contributions:** Conceptualization, M.Z., D.C. and Z.L.; methodology, M.Z., H.C. and D.P.; validation, D.P., F.P. and Z.L.; formal analysis, M.Z., D.P. and D.C.; investigation, M.Z. and D.C.; resources, M.Z., A.H., K.S. and Q.H.; data curation, M.Z., A.H., K.S. and Q.H.; writing—original draft preparation, M.Z.; writing—review and editing, M.Z., C.Y., Y.Z. and J.R.; visualization, M.Z.; supervision, H.C. and F.P.; project administration, Z.L. and D.C.; funding acquisition, M.Z., D.C. and Z.L. All authors have read and agreed to the published version of the manuscript.

**Funding:** This research was funded by the Natural Science Foundation Project of CQ CSTC, grant number cstc2017jcyjBX0022.

**Institutional Review Board Statement:** Not applicable.

**Informed Consent Statement:** Not applicable.

**Data Availability Statement:** Data sharing is not applicable.

**Conflicts of Interest:** The authors declare no conflict of interest.

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
