# Peer review of "Effects of Additives Containing Cyanopyridine on Electrodeposition of Bright Al Coatings from AlCl3-EMIC Ionic Liquids"

_coatings, doi:10.3390/coatings11111396_

Round 1
Reviewer 1 Report
Thanks for sharing your research.
The manuscript is well structured and clearly understandable. Conclusions are sound and supported by the measurements.
In order to improve the quality language revision is necessary, especially in terms of correct use of tense.
Just some examples from chapter 3.4:
line 314: change to "additives were added"
line 318: change to "know" or "have known"
These comments shall be considered when revising the whole manuscript.
Futhermore, fixed space (ctrl+alt+space) instead of just space shall be used between numpers and units to prevent unwanted line break between number and unit, which makes reading more difficult.
Author Response
To Reviewer 1

Reviewer 2 Report
- An important issue about this paper is that in the introduction section you need to describe in detail the application of these kinds of coatings and why you studied this kind of additive and also discuss about its toxicity. Is this coating corrosion protective for the copper substrate or this paper is just focused on the mechanism of the electrodeposition and the effect of different additives? It is also crucial to compare the cost of these kinds of coatings with different usual methods to apply aluminum coatings such as the hot-dip method. Some missed refs to be compared are: 1016/j.rser.2020.110100 , 10.1016/S1003-6326(20)65257-8 , 10.1016/j.surfcoat.2019.04.079 , 10.33003/fjs-2020-0402-190 , 10.1016/j.jallcom.2016.08.329 , 10.3390/coatings11010080 , 10.1016/j.surfcoat.2016.09.052 , 10.1016/j.matpr.2020.07.161
- If the application is for corrosion protection, in figure 9, if the corrosion tests you’ve done are for the corrosion of aluminum, then you should do the potentiodynamic tests. And if it is for the protection of the copper, aluminum is not the usual coating for these applications. So, you need to discuss them in detail.
Author Response
To Reviewer 2

Reviewer 3 Report
- The article focuses on the novel research work of developing additives as brightening agents and levelling agents.
- The authors needs to include SEM images highlighting the evolution of Al crystals with time. The authors have mentioned the deposition times upto 30 min but have not discussed the results anywhere in the manuscript. The images and corresponding effects needs to be discussed.
- Could the authors explain the reasons for choosing the concentration of additives to be 6 mM and deposition current density of 10 mA/ cm2 only? The influence of current density and the concentration of additives on the overall deposit needs to be discussed.
- It would be good to represent the texture coefficient of the deposits based on XRD and make an analysis
Author Response
To Reviewer 3

Round 2
Reviewer 2 Report
Accept